# MGL1 Receptor Plays a Key Role in the Control of *T. cruzi* Infection by Increasing Macrophage Activation through Modulation of ERK1/2, c-Jun, NF-κB and NLRP3 Pathways

**DOI:** 10.3390/cells9010108

**Published:** 2020-01-01

**Authors:** Tonathiu Rodriguez, Thalia Pacheco-Fernández, Alicia Vázquez-Mendoza, Oscar Nieto-Yañez, Imelda Juárez-Avelar, José L. Reyes, Luis I. Terrazas, Miriam Rodriguez-Sosa

**Affiliations:** 1Unidad de Biomedicina (UBIMED), Facultad de Estudios Superiores Iztacala (FES-Iztacala), Universidad Nacional Autónoma de México (UNAM). Tlalnepantla, Estado de México 54090, Mexico; tonathiurh@hotmail.com (T.R.); silaht16@gmail.com (T.P.-F.); biol.oscarnieto@gmail.com (O.N.-Y.); imelda_juarez@yahoo.com (I.J.-A.); jlreyes@iztacala.unam.mx (J.L.R.); literrazas@unam.mx (L.I.T.); 2Carrera de Optometría, FES-Iztacala, UNAM. Tlalnepantla, Estado de México 54090, Mexico; aliciavm@ired.unam.mx; 3Laboratorio Nacional en Salud, FES-Iztacala, UNAM. Tlalnepantla, Estado de México 54090, Mexico

**Keywords:** C-type lectin-like receptors, macrophage galactose-C type lectin, mouse MGL, *Trypanosoma cruzi*, PRRs, innate immunity response

## Abstract

Macrophage galactose-C type lectin (MGL)1 receptor is involved in the recognition of *Trypanosoma cruzi* (*T. cruzi*) parasites and is important for the modulation of the innate and adaptive immune responses. However, the mechanism by which MGL1 promotes resistance to *T. cruzi* remains unclear. Here, we show that MGL1 knockout macrophages (MGL1^−/−^ Mφ) infected in vitro with *T. cruzi* were heavily parasitized and showed decreased levels of reactive oxygen species (ROS), nitric oxide (NO), IL-12 and TNF-α compared to wild-type macrophages (WT Mφ). MGL1^−/−^ Mφ stimulated in vitro with *T. cruzi* antigen (*Tc*Ag) showed low expression of TLR-2, TLR-4 and MHC-II, which resulted in deficient splenic cell activation compared with similar co-cultured WT Mφ. Importantly, the activation of p-ERK1/2, p-c-Jun and p-NF-κB p65 were significantly reduced in MGL1^−/−^ Mφ exposed to *Tc*Ag. Similarly, procaspase 1, caspase 1 and NLRP3 inflammasome also displayed a reduced expression that was associated with low IL-β production. Our data reveal a previously unappreciated role for MGL1 in Mφ activation through the modulation of ERK1/2, c-Jun, NF-κB and NLRP3 signaling pathways, and to the development of protective innate immunity against experimental *T. cruzi* infection.

## 1. Introduction

The C-type lectins are a superfamily of more than 1000 proteins that are identified by having one or more characteristic C-type lectin-like domains (CTLDs) [1,2]. These molecules were originally named for their ability to bind carbohydrates in a calcium (Ca^+^)-dependent manner through conserved residues within the CTLDs. However, the CTLDs of many C-type lectins lack the components required for Ca^2+^-dependent carbohydrate recognition and can recognize a broader repertoire of ligands, including proteins, lipids and inorganic molecules [3].

In mammals, C-type lectins are found as secreted molecules or as transmembrane proteins known as transmembrane C-type lectin receptors (CLRs). Traditionally, it is widely accepted that CLRs that are expressed in antigen presenting cells (APCs) play an important role in the recognition, internalization of self (endogenous) and non-self (exogenous) ligands and regulate the routing of the internalized antigens to MHC-I or MHC-II-loading compartments for their further presentation to T cells, thereby creating a specific immune response [4,5]. However, it has recently been shown that CLRs also play an important role in promoting diverse physiological functions, such as the modulation of cellular, developmental, homeostatic and immunological responses [6,7].

Protozoa parasites possess numerous glycosylated structures that are capable of activating the innate immune response by binding to pattern recognition receptors (PRRs), such as toll like receptors (TLRs) and CLRs such as the mannose receptor (MR), dendritic cell-specific intercellular adhesion molecule 3-grabbing non-integrin (DC-SIGN), MGL and Dectin-1, among others [2,8]. Previous reports have shown that CLRs cooperate with TLRs to activate intracellular signaling pathways upon sensing pathogen-derived antigens [9,10,11]. For example, Dectin-1 and MR expressed in Mφ play a crucial role in the microbicidal response by inducing reactive oxygen species (ROS) production against the protozoan *Leishmania infantum* [12]. Despite their importance, many of the CLRs involved in cellular activities are poorly characterized, and their contribution and underlying mechanism remain incompletely understood.

Macrophage galactose-C type lectin is a CLR that is selectively expressed as a homo-oligomer on APCs such as immature dendritic cells (DC) and Mφ in humans and mice [11]. The carbohydrate recognition domain of MGL binds with high affinity to glyproteins expressing terminal galactose (Gal) and N-acetylgalactosamine (GalNAc) residues [13,14]. In mice, there are two homologs of human MGL, MGL1 and MGL2 [15]. MGL1 shares significant sequence homology with human MGL, recognizes terminal Gal and Lewis X structure residues and can mediate glycoprotein endocytosis [16,17], whereas MGL2 recognizes α-and β-GalNAc and does not interact with Lewis X structures [18]. The variation and distribution of MGL1 and MGL2 in healthy mouse cells shows a significant portion of the MGL1 single positive cells in bone marrow (BM), peritoneal and spleen cells. Specifically, a portion of conventional DC (cDC) co-express MGL1 and MGL2, while another portion of cDCs and plasmocytoid DCs (pDC) only express MGL1 (mostly cDC). The peritoneal exudate Mφ (PE-Mφ) and BMMφ expresses significant levels of MGL1, while MGL2 expression is low in PE-Mφ and almost absent in BMMφ [19]. 

The intracellular parasite *Trypanosoma cruzi* is the causative agent of Chagas’ disease, an important health problem in Latin America that is becoming an emerging global public health problem as a result of migration and global climate change [20]. Abundant amounts of mucin-like glycoproteins are present on this parasite’s surface. These glycoconjugates contain approximately 60% carbohydrates, most of which are anchored to glycosylphosphatidylinositol (GPI)- and glycoinositolphospholipid (GIPL)-mucin molecules [21]. We have previously reported that most *Tc*Ag bind to lectin Jacalin, which recognizes Gal residues; therefore, *Tc*Ag are glycosylated with Gal residues. In addition, we showed that MGL1^−/−^ mice were highly susceptible to *T. cruzi* infection, and they developed higher parasitemia and mortality rates than WT mice [22]. Since *Tc*Ag contains Gal residues and MGL1^−/−^ Mφ showed an impaired ability to eliminate this parasite, we hypothesized that MGL1 could play an important role in optimal Mφ activation. The mechanism by which the MGL1 receptor works has not been identified; therefore, we aimed to clarify the mechanism by which the MGL1 activates Mφ against *T. cruzi* infection. 

Here, we show that the absence of MGL1 led to impaired Mφ activation and we also provide additional evidence to support that MGL1^−/−^ Mφ had significantly reduced phosphorylation of subunit p65 of nuclear factor (p-NF)-κB, extracellular signal-regulated kinase 1/2 (p-ERK1/2) and transcription factor c-Jun (p-c-Jun), and decreased expression levels of the nucleotide-binding domain leucine-rich repeats family protein (NLRP3) in *T. cruzi* infection.

## 2. Materials and Methods

### 2.1. Mice

Six- to eight-week-old male MGL1^−/−^ mice on a C57BL/6 genetic background (donated by Glycomics Consortium, USA) were backcrossed for more than 10 generations [23]. WT C57BL/6 background mice were purchased from Harlan (Invigo, Mexico City, Mexico). Mice were maintained in a pathogen free environment at the FES-Iztacala, UNAM animal facilities. Genotyping of MGL1^−/−^ mice was routinely performed on DNA isolated from tail snips using a polymerase chain reaction (PCR) procedure [24]. The PCR were performed using the following primers: MGL1: forward 5′-CTTGGTCCCAGATCCGTATC-3′ and reverse 5′-ATGTCATGACTCAGGATC-3′; Neomycin (NEO): forward 5′-AGGATCTCCTGTCATCTCACCTTGCTCCTG-3′ and reverse 5′-AAGAACTCGTCAAGAAGGCGATAGAAGGCG-3′ (All synthesized by Sigma-Aldrich, Mexico City, Mexico). PCR for the amplification of MGL and NEO was performed with Taq DNA polymerase (Ampliqon, Bioreagents and Molecular Diagnostics) following the manufacturer’s instructions. A PCR fragment of 714 bp, corresponding to MGL, or 492 bp, corresponding to NEO, was visualized to identify WT or MGL1^−/−^ mice, respectively. The PCR products were analyzed by electrophoresis on a 1.5% agarose gel and were viewed under UV light (Bio-Rad, Mexico City, Mexico).

All experimental procedures using animals were designed to minimize suffering and the number of subjects used. These studies were conducted in accordance with the ethical standards approved and carried out under strict accordance with the guidelines for the Care and Use of Laboratory Animals adopted by the U.S. National Institutes of Health, and the Mexican Regulation of Animal Care and maintenance (NOM-062ZOO-1999, 2001). And it was revised and approved by the Ethics Committee at FES-Iztacala, UNAM (CE/FESI/062019/1311). 

### 2.2. Parasites

The Mexican *T. cruzi* TBAR/MX/0000/Queretaro strain belonging to DTU TcI was used in this work. Epimastigotes of *T. cruzi* were cultured at 28 °C in biphasic culture with brain heart infusion broth, agar and dextrose (Sigma-Aldrich, Mexico City, Mexico), and in the liquid phase with saline solution supplemented with 5% heat-inactivated fetal bovine serum (FBS, Thermo Fisher Scientific, Waltham, MA, USA) with 100 U of penicillin/streptomycin (all from GIBCO-BRL, Grand Island, NY, USA).

### 2.3. Soluble T. cruzi Lysate Antigen (TcAg)

Culture-derived epimastigotes were obtained and washed three times in phosphate buffered saline (PBS) by centrifugation at 1300× *g* for 10 min. The obtained pellet was sonicated six times for 10 s each at 50 W using a sonic Dimem-brator 300 (Thermo Fisher Scientific, Waltham, MA, USA) in the presence of protease inhibitors (Sigma-Aldrich). Parasite lysis was confirmed using a microscope. Parasite lysates were then centrifuged at 20,000× *g* for 30 min at 4 °C, and PBS-soluble antigens were stored at −70 °C until use. The protein concentration was determined using a Bradford protein kit (Sigma-Aldrich).

### 2.4. Cell preparations and T. cruzi Infection In Vitro

Peritoneal exudate cells (PECs) were obtained from the peritoneal cavity of MGL1^−/−^ and WT mice under sterile conditions using 10 mL of ice-cold Hank´s balanced salt solution (Microlab, Mexico City, Mexico). Following two washes with Hank’s balanced solution, red blood cells were lysed by resuspending the cells in Boyle’s solution (0.17 M Tris and 0.16 M ammonium chloride, all from Sigma-Aldrich). The viable cells were counted using the trypan blue exclusion method (routinely exceeding 95%) with a Neubauer hemocytometer (Sigma-Aldrich). PECs were adjusted to a concentration of 5 × 10^6^ cells/mL in Dulbecco’s modified Eagle’s medium (DMEM) supplemented with 10% FBS, 100 U of penicillin/streptomycin, and 2 mM glutamine (all from GIBCO-BRL), and cultured in 24-well plates (Costar, Bedford, MA, USA). After 2 h at 37 °C and 5% CO_2_, non-adherent cells were removed by washing with warm DMEM. Adherent cells (PE-Mφ) were removed from the plate by washing with 5 mM ethylenediaminetetraacetic acid (EDTA) in warm PBS and then adjusted to a concentration of 1 × 10^6^ cells/mL. PE-Mφ were allowed to adhere on coverslips in 24-well flat-bottomed culture plates for 2 h and then were infected with culture-derived epimastigotes at a ratio of 10:1 (parasites: Mφ) in supplemented DMEM. After 2 h of incubation, cells were washed with DMEM to remove the extracellular parasites; coverslips were removed, and stained with Giemsa (Sigma-Aldrich). A minimum of 100 cells in different microscopic fields were examined by a light Zeiss AXIO Vert A1 microscope (Carl Zeiss, Berlin, Germany), and the percentage of infected cells and the number of amastigotes per infected cell were determined in triplicate in a double-blind manner. Five independent experiments were analyzed. Supernatants were collected and stored at −70 °C until use for the quantification of NO and cytokine production (IL-1β, IL-10, IL-12 and TNF-α).

### 2.5. Reactive Oxygen Species Activity Assay

PE-Mφ from MGL1^−/−^ and WT mice grown on coverslips as mentioned above were exposed to culture-derived epimastigotes at a ratio of 10:1 (parasites: 1 Mφ) or *Candida albicans* (*C. albicans*) (5 × 10^6^ cells/mL) for 2 h at 37 °C and 5% CO_2_. Next, ROS production was measured by the nitrobluetetrazolium (NBT) test, by adding 100 µL of 0.1% NBT (Sigma-Aldrich) to a final volume of 1 mL per well with DMEM. The resting plate was incubated at 37 °C for 1 h. The coverslips were rinsed with saline to wash away excess NBT. Mφ were stained with 0.5% safranin for 7 min and washed with distilled water. Finally, the coverslips were mounted on glass slides with resin, and 100 cells in different fields were examined under a light microscope (Zeiss AXIO Vert A1 microscope; Carl Zeiss, Berlin, Germany); the percentage of ROS positive cells was determined in a double-blinded fashion.

### 2.6. Detection of Nitric Oxide and Cytokines Production

PE-Mφ from MGL1^−/−^ and WT mice were grown in 24-well flat-bottomed plate, as mentioned above, were left untreated or treated with lipopolysaccharide (LPS; 100 ng/mL, *Escherichia coli* 0111:B4, Sigma-Aldrich), IFN-γ (10 U/mL, Peprotech, Mexico City, Mexico), or LPS+IFN-γ (100 ng/mL + 10 U/mL) for 24 h. Then, they were infected with culture-derived epimastigotes at a ratio of 10:1 (parasites: Mφ) for 2 h at 37 °C and 5% CO_2_. Cells were washed with DMEM to remove the extracellular parasites, and the plates were incubated for an additional 24 h. Supernatant from MGL1^−/−^ PE-Mφ and WT PE-Mφ cultures, were assayed for nitric oxide (NO) production by the Griess reaction adapted for 96-well plates (Costar) [25]. Briefly, 50 µL of culture supernatant was mixed with an equal volume of Griess reagent and incubated for 10 min at room temperature in the dark. The concentration was determined by extrapolating the optical density from each sample to a standard curve of sodium nitrite. The absorbance was measured at 570 nm in Epoch microplate spectrophotometer (BioTEk, Winooski, VT, USA). To determine IL-12, TNF-α, IL-10 and IL-1β levels by using commercially available enzyme-linked immunosorbent assays (ELISAs) according to the manufacturer’s instructions (Peprotech, Mexico City, Mexico). The optical density (OD) was measured using an Epoch microplate spectrophotometer (BioTEk) at 405 nm.

### 2.7. Viability of Internalized T. cruzi

To test the trypanocidal capacity of both MGL1^−/−^ and WT Mφ, parasite proliferation was examined as previously described [26]. Briefly, Mφ from MGL1^−/−^ and WT mice were obtained as described in Section 2.4 and adjusted to 1 × 10^6^ cells/mL. These cells were seeded in 24-well flat-bottom culture plates, and incubated for 2 h at 37 °C and 5% CO_2_. Mφ were washed with DMEM, and divided into groups as follows: non-stimulated or stimulated with LPS (100 ng/mL), IFN-γ (10 U/mL) or LPS+IFN-γ (100 ng/mL + 10 U/mL) for 24 h. Then, Mφ were infected with culture-derived epimastigotes at a ratio of 10:1 (parasites: Mφ) in supplemented DMEM. After 2 h, Mφ were washed twice with DMEM to remove non-internalized parasites, and fresh supplemented DMEM was added to the culture and Mφ were incubated for another 24h. After this time supernatants were collected and stored at −70 °C until use for NO quantification and cytokine production (IL-1β, IL-10, IL-12 and TNF-α). Mφ were lysed using 0.01% sodium dodecyl sulfate (SDS) in 100 mL of warm PBS for 30 min and pipetted up and down 5–10 times. Released amastigotes from the Mφ were harvested and centrifuged at 1300× *g*. The pellet was resuspended in 600 μL of supplemented DMEM medium, and aliquots (150 μL) of suspension were seeded into 96-well flat-bottom culture plates (Costar) at 37 °C and 5% CO_2_ for 72 h. Eighteen hours prior to culture termination, 0.5 µCi of tritiated thymidine ([^3^H]TdR, 185 GBb/mmol activity; Amersham, Aylesbury, UK) was added to each well. The cells were harvested on fiberglass paper (PerkinElmer, Billerica, MA, USA), and the counts per minute (CPM) were quantified using a liquid scintillation counter Trilux 1450 Microbeta (Tomtec, Hamden, CT, USA). The Mφ trypanocidal activity was measured as a reduction in the incorporation of [^3^H]TdR by surviving amastigotes recovered from the Mφ that become into epimastigotes in the cell-free medium.

### 2.8. Flow Cytometry Analysis 

PE-Mφ from MGL1^−/−^ and WT mice were obtained and stimulated with *Tc*Ag (25 µg/mL) for 24 or 48 h. These cells were incubated with anti-mouse FcγR antibody (CD16/CD32) in staining buffer (1× PBS, 2% FBS, 1% NaN3) for 15 m, followed by incubation for 30 m at 4 °C with FITC-conjugated anti-F4/80, PE-conjugated anti-MGL1 and APC-conjugated anti-MGL2 antibodies (BioLegend, SD, CA). For quantification of costimulatory molecule expression, MGL1^−/−^ and WT PE-Mφ and BMMφ were stimulated in vitro with LPS (100 ng/mL) or *Tc*Ag (25 µg/mL) for 24 h or infected with culture-derived epimastigotes for 2 h (ratio 1:10). The cells were incubated with the following fluorochrome-conjugated Abs: Pacific blue anti-F4/80, PerCP/Cy5.5 anti-CD11b, PE anti-TLR-4, PE anti-MHC-II, FITC anti-TLR-2, FITC anti-CD40 and FITC anti-CD80 (all from BioLegend), as well as the negative control. Mφ were washed three times with FACS buffer and fixed in 0.8% paraformaldehyde before acquisition and analysis (Attune NxT, Thermo Fisher Scientific, Waltham, MA, USA).

### 2.9. Co-culture of Mφ and Splenic Cells

To examine the antigen presenting capacity of MGL1^−/−^ Mφ, Mφ were co-cultured with splenocytes as follows: PE-Mφ from MGL1^−/−^ or WT mice were obtained and adjusted to 1 × 10^6^ cells/mL as described above. PE-Mφ were seeded (100 µL) in 96-well flat-bottom culture plates (Costar) and stimulated with 100 µL of *Tc*Ag (25 µg/mL) at 37 °C and 5% CO_2_. Two hours later, PE-Mφ were washed three times to remove the non-phagocytosed antigen. Splenocytes from WT mice infected with *T. cruzi* for 21 days were added at a ratio of 1:10 (Mφ: splenocytes; 100 µL of splenocytes adjusted to a concentration of 10 × 10^6^ cells/mL). Co-cultures were maintained at 37 °C and 5% CO_2_ for 72 h, and then 0.5 µCi/well of [^3^H]-thymidine (185 GBb/mmol activity; Amersham, Aylesbury, UK) was added and incubated for additional 18 h. The plate was harvested on fiberglass paper (PerkinElmer) using a 96-well harvester (Tomtec, Toku, Finland), and CPM were quantified using a liquid scintillation Trilux 1450 Microbeta counter (Tomtec). 

### 2.10. Protein Levels of NF-κB, P38, ERK1/2 and NLRP3 Signaling Pathways Detected by Western Blotting

Murine BMMφ were generated using tibias and femurs aseptically removed from MGL1^−/−^ and WT mice as previously described [27]. Briefly, bone ends were cut and flushed with 10 mL of sterile PBS. The obtained cell suspension was centrifuged at 1300× *g* for 10 min at 4 °C. Cells were adjusted to a concentration of 4 × 10^6^ cells/mL in Mφ differentiating medium containing supplemented DMEM (20% FBS) and 50 ng/mL murine macrophage colony-stimulating factor (M-CSF) (Biotech, BG, DE). Two milliliters of cell suspension was seeded into each well of a 6-well plate and incubated at 37 °C in 5% CO_2_. After 72 h, 1 mL of differentiating medium was added to each well. Cells were allowed to differentiate for 7 days. BMMφ were washed twice, adjusted to a concentration of 4 × 10^6^ cells/mL and stimulated with LPS (100 ng/mL) or *Tc*Ag (25 μg/mL) for 0, 5, 15 and 30 min. BMMφ protein was extracted using Laemmli buffer (containing, 92 mM Tris (pH 6.8), 18% glycerol, 1.8% SDS, 0.02% bromophenol blue and 2% β-mercaptoethanol (all from Sigma-Aldrich) with protease and phosphatase inhibitors (Roche Diagnostic, Basel Switzerland) according to the manufacturer’s instructions. The samples were centrifuged at 700× *g* for 5 min at 4 °C, and the protein concentration was determined using a Bradford assay (Sigma-Aldrich). Protein samples (15 µg) were separated by 12% sodium dodecyl sulfate-polyacrylamide (SDS-PAGE) gel electrophoresis at 80 V and were transferred to immobilon-P membranes (0.22 μM, Millipore, Bedford, MA, USA) by electroblotting. The membranes were blocked for 2 h at room temperature in Tris-buffered saline-Tween 20 (TBST) supplemented with 5% *w/v* bovine serum albumin (Sigma-Aldrich). Subsequently, the membrane was washed three times with TBST and incubated at 4 °C overnight on a shaker plate with the following primary antibodies: GAPDH (as housekeeping protein), NF-κB p65, p-NF-κB p65, p38 MAPK, p-p38 MAPK, p44/42 MAPK (ERK1/2), p-p44/42 MAPK (p-ERK1/2) and NLRP3 following the manufacturer’s protocol (Cell Signaling, Danvers, MA, USA). After washing the membrane with TBST four times, an alkaline phosphatase-conjugated secondary antibody in TBST was added (dilution 1:5000; Cell Signaling) and incubated for 2 h at room temperature. The membrane was washed with TBST four times, the signal was revealed using Super Signal West Femto (Thermo Fisher Scientific) and then scanned and analyzed using a fluorescent Odyssey infrared scanner (LI-COR, Lincoln, NE, USA). 

### 2.11. Statistical Analysis 

Comparisons between the WT and MGL1^−/−^ groups were made using the unpaired Student’s t-test or one-way ANOVA followed by Dunnett’s multiple comparisons test using GraphPad Prism Program version 6.0 (GraphPad Software, La Jolla, CA, USA). All data are shown as the mean ± standard error of the mean (SEM) for at least two or three independent experiments. A value of *p* < 0.05 was considered significant.

## 3. Results

### 3.1. Trypanosoma cruzi Antigens Induce High Expression of MGL1 and Moderate Expression of MGL2 in PE-Mφ

To verify the expression of MGL1 and MGL2 receptors, PE-Mφ (F4/80^+^) from WT and MGL1^−/−^ mice were analyzed by flow cytometry after stimulation in vitro for 24 and 48 h with *Tc*Ag. As Denda-Nagai et al. reported [19], WT PE-Mφ showed surface expression of the MGL1 and no expression of MGL2 receptors at baseline (Figure 1a,b; WT-Mφ, non-stimulated, gray shadow). After stimulation with *Tc*Ag for 24 or 48 h, MGL1 expression increased significantly while MGL2 expression increased moderately (Figure 1a,b; WT-Mφ, gray shadow). These observations indicate that MGL1 receptor is highly expressed, and it is mostly induced by the *Tc*Ag in WT PE-Mφ, compared to MGL2 expression.

PE-Mφ from MGL1^−/−^ mice did not show expression of MGL1, but moderate levels of MGL2 expression were observed at baseline (Figure 1a,b; MGL1^−/−^ Mφ, non-stimulated, solid line). Importantly, after *Tc*Ag stimulation, MGL1^−/−^ PE-Mφ did not show expression of MGL1, whereas MGL2 was slightly elevated (Figure 1a,b; MGL1^−/−^ Mφ, solid line). These results demonstrate that PE-Mφ from MGL1^−/−^ mice are unable to upregulate significantly MGL2 in response to *Tc*Ag.

### 3.2. The MGL1 Receptor Plays a Role in Controlling T. cruzi PE-Mφ Infection

Pathogens can be recognized by CLRs, and some receptors play a major role in pathogen internalization. We previously demonstrated that MGL1 binds *Tc*Ag, and MGL1^−/−^ mice were found to be more susceptible to in vivo *T. cruzi* infection [2]. Herein, we explored whether the MGL1 receptor plays a role in vitro internalization of *T. cruzi* parasites by PE-Mφ. To address this question, PE-Mφ from MGL1^−/−^ and WT mice were infected with culture-derived epimastigotes of *T. cruzi.* The percentage of infected Mφ was determined 2 h after co-incubation. We observed that MGL1^−/−^ Mφ exhibited a higher percentage of infected Mφ (86%) than WT Mφ (48%) (Figure 2a, *p* < 0.05). Importantly, MGL1^−/−^ Mφ also exhibited a higher number of internalized parasites per cell. Specifically, 33% of MGL1^−/−^ Mφ showed 2–3 amastigotes per Mφ, while 14% of WT Mφ showed 2–3 amastigotes per Mφ (Table 1, *p* < 0.05; and Figure 2b). 

As LPS/IFN-γ-treatment induces classically activated Mφ, which are involved in the control of *T. cruzi* infection, we asked whether pre-activated MGL1^−/−^ Mφ could control *T. cruzi* infection. To address this question, culture-derived epimastigotes of *T. cruzi* were used to infect untreated MGL1^−/−^ and WT Mφ or treated with IFN-γ, LPS and LPS/IFN-γ for 24 h. Parasite survival after 24 h was determined by measuring [^3^H]thymidine incorporation as a marker of parasite proliferation. Untreated WT PE-Mφ displayed greater [^3^H]thymidine uptake, indicating parasite proliferation (Figure 2c). As expected, treatment with IFN-γ, LPS or both, IFN-γ and LPS resulted in a significant reduction in parasite survival, as reduced [^3^H]thymidine uptake was observed in this WT PE-Mφ (Figure 2c). Interestingly, untreated MGL1^−/−^ PE-Mφ harbored more surviving *T. cruzi* parasites than untreated WT PE-Mφ, as shown by the significantly increased uptake of [^3^H]thymidine (*p* < 0.05%). Despite being treated with IFN-γ, LPS or both, MGL1^−/−^ PE-Mφ showed higher levels of parasite proliferation compared with WT PE-Mφ. MGL1^−/−^ PE-Mφ therefore displayed an impaired ability to eliminate these parasites, as they maintained a higher [^3^H]thymidine uptake than their WT PE-Mφ counterparts. These results suggest that MGL1^−/−^ PE-Mφ have impaired trypanocidal ability compared to WT PE-Mφ.

### 3.3. MGL1^−/−^ PE-Mφ Have a Deficient Oxidative Burst, as well as Nitric Oxide and Proinflammatory Cytokine Production during T. cruzi Infection

Reactive oxygen species (ROS) are reactive molecules that include oxygen ions, free radicals and peroxides. ROS are produced as a result of the enzymatic activity that is acquired by the phagosome during its formation. Indeed, the superoxide burst in Mφ against intracellular parasites represents one of the main antimicrobial mechanisms involved in host defense [28]. To determine whether the MGL1 receptor was associated with ROS production, we performed the NTB test in MGL1^−/−^ and WT Mφ infected for 2 h with culture-derived epimastigotes of *T. cruzi* or *C. albicans* as a positive control. 

We observed a similar increase in ROS production in both MGL1^−/−^ and WT PE-Mφ infected with *C. albicans*. However, when PE-Mφ from WT mice were infected with *T. cruzi*, they displayed a significant increase in ROS production, whereas *T. cruzi*-infected Mφ from MGL1^−/−^ mice did not (Figure 3a). 

In addition to ROS production, Mφ also produce reactive nitrogen species, specially nitric oxide (NO), that, together with IL-12 and TNF-α, plays an important role in controlling the initial infection by *T. cruzi* [21,29]; in contrast, the anti-inflammatory cytokine IL-10 is associated with susceptibility in *T. cruzi* infection [30]. Thus, we tested NO and cytokine production in MGL1^−/−^ and WT PE-Mφ treated with LPS, IFN-γ or LPS/IFN-γ for 24 h. These primed PE-Mφ were later infected with epimastigotes of *T. cruzi* for 2 h, they were washed to remove the extracellular parasites and incubated for an additional 24 h. Supernatants were collected for NO and cytokine quantification. 

As expected, WT PE-Mφ produced slightly elevated NO after IFN-γ stimulation, moderate levels of NO in response to LPS and high levels of NO production in response to LPS/IFN-γ with or without *T. cruzi* infection. However, the MGL1^−/−^ PE-Mφ did not produce NO in response to *T. cruzi* infection. Indeed, we observed that MGL1^−/−^ PE-Mφ stimulated with LPS, or LPS/IFN-γ displayed a deficient production of NO with or without *T. cruzi* infection compared with WT PE-Mφ (Figure 3b).

The proinflammatory cytokines such as IL-12 and TNF-α were induced in response to IFN-γ, LPS or LPS/IFN-γ in WT PE-Mφ without or with *T. cruzi* infection. Non infected PE-Mφ from MGL1^−/−^ mice showed similar levels of IL-12 and TNF-α in response to IFN-γ stimulus. Interestingly, the same MGL1^−/−^ PE-Mφ displayed decreased production of IL-12 and TNF-α in response to LPS. Upon *T. cruzi* infection, MGL1^−/−^ PE-Mφ stimulated with IFN-γ showed a decreased production of IL-12, and similar levels of TNF-α compared with WT PE-Mφ (Figure 3c).

IL-10 production showed similar levels in non-infected WT and MGL1^−/−^ PE-Mφ supernatants, after treatment with IFN-γ, LPS or LPS/IFN-γ. The production of IL-10 increased slightly in WT PE-Mφ only with infection or with infection + IFN-γ. However, no induction of IL-10 was detected in the MGL^−/−^ PE-Mφ in *T. cruzi* infection, or with any other stimulus (Figure 3c). These results together support the hypothesis that MGL1 favors the production of ROS, NO and inflammatory cytokines during *T. cruzi* infection.

### 3.4. MGL1^−/−^ PE-Mφ Exhibit Deficient Activation of T. cruzi Antigen-Specific Lymphocytes

Next, we analyzed the expression of TLR-2, TLR-4, MHC-II, CD40 and CD80 in MGL1^−/−^ and WT PE-Mφ stimulated in vitro with LPS or *Tc*Ag or infected with epimastigotes of *T. cruzi* parasites (Figure 4). 

Consistent with previous reports that demonstrated that MGL can modulate TLR activation [22,31], we observed significantly lower expression levels of TLR-2 and TLR-4 in MGL1^−/−^ PE-Mφ compared to WT PE-Mφ treated with *Tc*Ag or *T. cruzi* parasites (Figure 4a,b). Importantly, we also found reduced expression of MHC-II and CD40 in MGL1^−/−^ Mφ (Figure 4c,d). However, no significant differences were observed in other costimulatory molecules such as CD80, between MGL1^−/−^ and WT Mφ treated with *Tc*Ag or *T. cruzi* parasites (Figure 4e). We hypothesized that the deficiency of these molecules could impair the performance of Mφ as antigen-presenting cells and decrease *T. cruzi*-specific lymphocyte activation. Thus, PE-Mφ from WT or MGL1^−/−^ mice were pre-loaded with *Tc*Ag to activate *T. cruzi*-specific splenocytes (coming from sensitized mice) in co-cultures. Splenocytes co-cultured with MGL1^−/−^ PE-Mφ displayed significantly lower proliferation than splenocytes co-cultured with WT PE-Mφ (Figure 5). These results suggest that lack of MGL1 may also modulate the adaptive response against *T. cruzi*.

### 3.5. Trypanosoma cruzi Antigens Induce High Expression of MGL1 and Low Expression (Almost Absent) of MGL2 in BMMφ 

It was previously reported that BMMφ express significant levels of MGL1, while MGL2 is absent in these cells. Therefore, to determine whether the recognition of *Tc*Ag by MGL1 could mediate the activation of Mφ, we decided to use BMMφ to establish a possible route of activation.

We first, search for the expression of MGL1 and MGL2 in WT BMMφ non-stimulated or stimulated with *Tc*Ag by 24 or 48 h. WT BMMφ showed expression of the MGL1 and almost absent expression of MGL2 at baseline (Figure 6a,b; WT-Mφ, non-stimulated, gray shadow). After stimulation with *Tc*Ag, MGL1 expression increased significantly, while MGL2 expression did not (Figure 6a,b; WT-Mφ, gray shadow). These observations suggest that *Tc*Ag may be recognized by MGL1 and favors its up-regulation, whereas MGL2 seems does not participate in this interaction.

Next, the MGL1 and MGL2 expression was examined using BMMφ from MGL1^−/−^ mice. As show in Figure 6, the absence of expression of MGL1 and MGL2 was confirmed on BMMφ from MGL1^−/−^ mice at baseline (Figure 6a,b; MGL1^−/−^ Mφ, non-stimulated, solid line). After *Tc*Ag stimulation, MGL1^−/−^ BMMφ did not show expression of MGL1, neither MGL2 (Figure 1a,b; MGL1^−/−^ Mφ, solid line). These results demonstrate that BMMφ from MGL1^−/−^ mice are unable to upregulate MGL2 in response to *Tc*Ag, suggesting that there is not a compensatory hyperexpression of this CLR in the absence of MGL1.

### 3.6. MGL1 Deficiency in BMMφ Results in Reduced Activation of the NFκ-B and ERK1/2 Signaling Pathways in Response to T. cruzi Antigen

The activation of Mφ by infectious and non-infectious insults is usually triggered by the recognition of the antigen by receptors on the surface of Mφ; this leads to the activation of the NFκ-B and c-Jun transcription factors, which play key roles in modulating the expression of many proinflammatory genes and innate immune response signaling [32,33,34]. Consequently, IL-10, TNF-α, and NO production by Mφ correlates with the strength of NFκ-B activation. Moreover, it is well established that the MAPK cascade also controls the post-transcriptional regulation of TNF-α [35]; the kinases ERK and P38 are notable members of the MAPK family. Therefore, to investigate the role of MGL1 in the activation of ERK1/2, p38, c-Jun and NFκ-B; WT and MGL1^−/−^ BMMφ were stimulated for 5, 15 and 30 min with LPS or *Tc*Ag, and the phosphorylation of these proteins was measured.

MGL1^−/−^ Mφ showed a significant reduction in the phosphorylation of ERK1, 2, c-Jun and NFκ-B after LPS or *Tc*Ag stimulation compared with that of WT Mφ. In contrast, comparable levels of phosphorylated p38 were observed between WT and MGL1^−/−^ Mφ (Figure 7a,b). These results suggest that the MGL1 may be signaling trough the ERK1/2 c-Jun and NFκ-B axis in response to *Tc*Ag, but not p38, thus MGL1 is critical for the optimal activation of Mφ during *T. cruzi* infection and consequently favor secretion of proinflammatory cytokines.

### 3.7. MGL1 Regulates the Expression of the NLP3 Sensor

One of the main factors that trigger acute or chronic inflammation is the inflammasome [36]. Assembly of PRRs-mediated inflammasomes, such as the NLRP3 sensor, one of the most important inflammasomes involved in the development of the inflammatory microenvironment through the activation of caspase-1, results in the secretion of IL-1β, which ultimately creates an inflammatory microenvironment [37]. Because MGL1^−/−^ Mφ exposed to LPS or *Tc*Ag exhibited decreased production of proinflammatory cytokines, which also correlated with deficient ERK1/2, c-Jun and NFκ-B activation, we explored whether MGL1^−/−^ Mφ may display deficient expression of the NLRP3 receptor. WT and MGL1^−/−^ BMMφ were stimulated with LPS, *Tc*Ag or LPS/*Tc*Ag for 24 h. After incubation, total protein was extracted, and western blot analysis was performed to quantify protein expression levels of NLRP3, procaspase-1 and caspase-1.

As shown in Figure 8a,b; WT BMMφ stimulated with LPS, *Tc*Ag or LPS/*Tc*Ag displayed increased NLRP3 protein expression levels compared to non-stimulated cells. Importantly, compared to WT BMMφ, MGL1^−/−^ BMMφ did not show any increase in NLRP3 protein levels neither procaspase-1, or caspase-1 upon stimulation with LPS/*Tc*Ag or either stimulus alone. Interestingly, MGL1^−/−^ BMMφ also displayed reduced IL-1β production in response to LPS, *Tc*Ag or LPS/*Tc*Ag compared to those levels found in WT BMMφ exposed to the same stimuli. 

## 4. Discussion

Recent evidence suggests that MGL plays an important role that goes beyond the sensing and elimination of dead and dying cells [38]. For example, MGL is able to recognize abnormal glycosylation patterns, a common characteristic of malignant cells, which allows the interaction of cancer cells with platelets, leukocytes and endothelial cells, facilitating tumor invasion, metastasis and the immunosuppressive response [39]. Therefore, GalNAc-carrying tumor-associated antigens or anti-MGL antibodies have been used as ligands to identify the role of MGL in the activation of DCs in vitro [19,31,40,41,42]. 

The generation of MGL1^−/−^ mice allows for investigating the immunological functions of MGL1 separately from that of MGL2 at the cellular level; using these mice, it has been possible to establish that MGL1 is capable of recognize glycosylated structures expressed in parasites including protozoa and helminths [8,22]. Our previous studies, conducted in a mouse model, demonstrated that the interaction of MGL1 with highly glycosylated structures of *T. cruzi* (as a natural ligand) plays an essential role in immunity by increasing Mφ activation and parasite killing during in vivo *T. cruzi* infection [22,43]. Despite this evidence pointing to a role for MGL1 in the immune response, how MGL1 modulates the immune response against this pathogen remains unclear. In the current study we demonstrated that in PE-Mφ and BMMφ from MGL1^−/−^ mice, MGL1 is totally absent, while the MGL2 expression is very low or absent, respectively. Thus, using MGL1^−/−^ BMMφ, we provide important insights into the mechanism by which the MGL1-*Tc*Ag interaction activates Mφ against *T. cruzi* infection. 

Having demonstrated that MGL1^−/−^ Mφ infected in vitro with *T. cruzi* presented greater numbers of internalized parasites than WT Mφ [22], we extended this observation by showing that internalized parasites in MGL1^−/−^ Mφ remain alive. Moreover, following overnight incubation of MGL1^−/−^ Mφ with IFN-γ, LPS or both, there was not a significant reduction in the number of intracellular parasites, as occurred in the WT Mφ. This activation deficiency of MGL1^−/−^ Mφ was accompanied by a remarkable downregulation of ROS and NO production, as well as by a reduction in the levels of IL-12 and TNF-α, which are important for controlling parasite infection and replication [44,45]. These observations suggest that MGL1 may recognize *T. cruzi* and favors a cross-talk between MGL1 signalling pathway and other important pathways for Mφ activation that helps to activate the oxidative burst. 

We previously observed that MGL1^−/−^ Mφ infected in vitro with trypomastigotes of *T. cruzi* (extracellular and infective blood form) had a negatively regulated expression of MHC-II and TLR4 [22]. We asked whether this phenomenon also occurred in MGL1^−/−^ Mφ treated with *Tc*Ag or epimastigotes of *T. cruzi* (a form of transition found in crops and in the insect vector). We decided to expose Mφ to culture-derived epimastigotes and not to blood trypomastigotes due to the fact that this later stage could be opsonized and major antigens neutralized [46]. Moreover, *T. cruzi* epimastigotes share GPI-anchored antigens with trypomastigotes [47], and we have previously shown that epimastigotes express MGL1 ligands (Galactose and N-Acetylgalactosamine) as revealed by staining with the lectin Jacalin [22]. Here, we demonstrate that MGL1^−/−^ Mφ exposed to *Tc*Ag or live epimastigotes of *T. cruzi* displayed downregulated expression of TLR4, MHC-II and CD40, but not CD80 expression, which correlated with a deficiency in *Tc*Ag presentation to activate antigen-specific T cells. This finding is consistent with a previous report by Napoletano et al., who showed that the interaction of MGL with tumor-associated antigens or an anti-MGL antibody in DCs improved the performance of DCs as antigen-presenting cells, promoting the positive regulation of markers of maturation and increasing the activation of antigen-specific CD8 T cells [41]. In contrast, van Vliet et al. observed that anti-MGL antibody treatment combined with TLR stimulation did not affect the expression of CD80, CD83, CD86 or MHC-II [31]. This discrepancy may suggest that the expression of costimulatory molecules, driven by MGL, is different depending on the stimulus. For instance, the MGL1^−/−^ PE-Mφ infected with the fungi *C. albicans* displayed similar ROS production compared to WT PE-Mφ, indicating a MGL1-independent recognition of this pathogen as well as an efficient oxidative burst. In line with this idea, our previous report demonstrated that MGL1^-/ -^PE-Mφ exposed to POLY: IC, a TLR-4-independent stimulus, induced high levels of NO and inflammatory cytokines [22]; indicating that Mφ activation is intact if the insult is independent of the interaction with MGL1, TLR4 or both.

*T. cruzi* parasites have abundant amounts of glycosylated molecules such as GPI- and GIPL-mucins, which are potent activators of TLR-2 and TLR-4, respectively [48]. Although Mφ stimulated with GPI-mucin were found to produce TNF-α, IL-12 and NO via TLR-2 [49], Mφ from TLR-2^−/−^ mice displayed partial phosphorylation of ERK1/2, detectable levels of TNF-α and reactive nitrogen intermediate production, compared to the completely unresponsive Myd88^−/−^ (myeloid differentiation factor 88; an essential signal transducer for TLRs) Mφ, indicating that *T. cruzi* parasites activate an alternative inflammatory pathway independent of TLR-2 [44]. Monocytes stimulated with GIPL-mucin induce NFκ-B activation via TLR-4, whereas TLR-4/MD-2 CHO cells (TLR-4 non-functional monocytes) do not [50]. Importantly, in DCs the costimulation of MGL with agonistic antibodies or carbohydrate ligands augments TLR-2-mediated responses, favoring IL-10 and TNF-α secretion [31]. Here, we showed that MGL1^−/−^ Mφ infected in vitro with epimastigotes of *T. cruzi* or exposed to *Tc*Ag showed reduced expression of TLR-2 and TLR-4, consequently, MGL1^−/−^ Mφ displayed a poor response to LPS. This support that MGL1, in coordination with TLR-4, is involved in *T. cruzi* recognition and Mφ activation, which could explain the defective production of TNF-α and IL-12, as well as the reduced ROS and NO production observed in MGL1^−/−^ Mφ.

Furthermore, we found reduced phosphorylation of ERK1/2 and NFκ-B in MGL1^−/−^ Mφ stimulated with *Tc*Ag compared to similarly exposed WT Mφ. These observations are consistent with previous reports demonstrating that MGL engagement with GalNac-carrying tumor-associated antigens or anti-MGL antibodies induced the phosphorylation of ERK1/2, c-Jun and NFκ-B activation in human DCs [31,41]. Thus, our results reveal that the possible recognition of *T. cruzi* by MGL1 activates intracellular signaling cascades that modulate the innate immune response, and this role requires the simultaneous activation of TLR molecules for its immune effects.

Although it is well known that TLRs profoundly influence the Mφ inflammatory response, this is not the only way by which the inflammatory response can be activated. Studies have shown that, through both direct and indirect mechanisms, CLRs such as Galectin-3 and Dectin-1 can activate the NLRP3, NLRC4 or caspase-8 inflammasomes, leading to the production of IL-1β [51,52]. Our study provides evidence for the first time that the MGL1 receptor contributes to NLRP3 inflammasome expression in response to both LPS and *Tc*Ag. 

Inflammasome activation is crucial for host defence in the acute phase of *T. cruzi* infection. In a previous report, mice lacking NLRP3 or caspase-12 genes exhibited increased numbers of *T. cruzi* parasites [53]. Moreover, *Tc*Ag or LPS increase NLRP3 activation in Mφ from WT mice [54]. However, we observed that MGL1^−/−^ Mφ had a diminished ability to produce IL-1β most likely as a result of the impaired inflammasome activation, in response to LPS and *Tc*Ag. In *T. cruzi* infection, the processing of IL-1β into its bioactive form requires two stimuli: first, the recognition of the pathogen that induces pro-IL-1β gene transcription, then ROS elicited by *T. cruzi* infection serve as the second signal for caspase-1 induction that triggers pro-IL-1β cleavage to active IL-1β [54,55]. It seems that the lack of interaction between *Tc*Ag and MGL1 in our model inhibits the expression of the inflammasome NLRP3, consequently, also the canonical pathway of caspase-1. This suggests that overall the inflammasome machinery is dampened in MGL1-deficient Mφ, probably because *T. cruzi* is not fully recognized in the absence of MGL1. We detected some IL-1β, even in the absence of ROS, suggesting that there is a compensatory mechanism or NLRP3-independent release of IL-1β bioactive, which could be an interesting topic for a future study. Whether the non-canonical (caspase 11-mediated) NLRP3 inflammasome pathway is altered in MGL1^−/−^ Mφ remains to be confirmed. Our findings are in line with those previously reported showing that other lectins, such as galectin-3, are able to trigger NLRP3 activation in the context of liver diseases and influenza infection [52,56]. 

Based on the results generated in this work, we propose a hypothetical model to explain the role of MGL1 in Mφ activation, in response to *T. cruzi* infection (Figure 9).

## 5. Conclusions

In conclusion, here, we showed that MGL1 engagement with *Tc*Ag enables Mφ to perform APC functions and provide additional evidence to support the hypothesis that MGL1 can upregulate Mφ activation. Another novel finding was that MGL1 synergizes with TLR-2 and TLR-4 to upregulate ERK1/2, c-Jun, NFκ-B-dependent expression of proinflammatory factors. Furthermore, we provided the first evidence that MGL1 contributes to the innate immune response via the NLRP3 inflammasome expression and ROS production in response to *Tc*Ag and LPS. Finally, our data offer insight into the mechanisms involved in controlling *T. cruzi* infection by MGL1, although more details of this early response have yet to be described.

## Figures and Tables

**Figure 1 cells-09-00108-f001:**
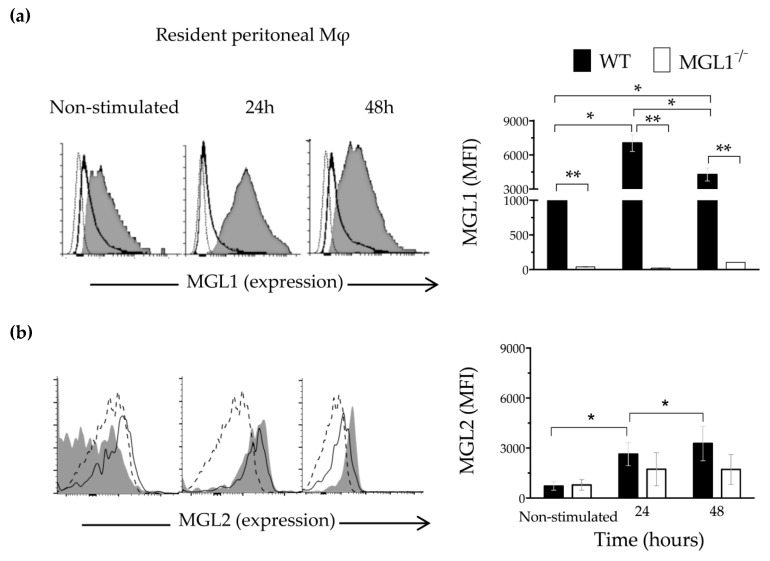
Expression of MGL1 and MGL2 in Mφ in response to *T. cruzi* antigen. PE-Mφ (F4/80+) from WT or MGL1^−/−^ mice were stimulated for 24 or 48 h with *Tc*Ag (25 µg/mL). Representative histogram and bar chart of the percentage of PE-Mφ expressing MGL1 and MGL2 are shown in (**a**) and (**b**), respectively. Dotted line, isotype; gray area, PE-Mφ from WT mice; solid line, PE-Mφ from MGL1^−/−^ mice; n = 6 mice per group; * *p* < 0.05 and ** *p* < 0.002.

**Figure 2 cells-09-00108-f002:**
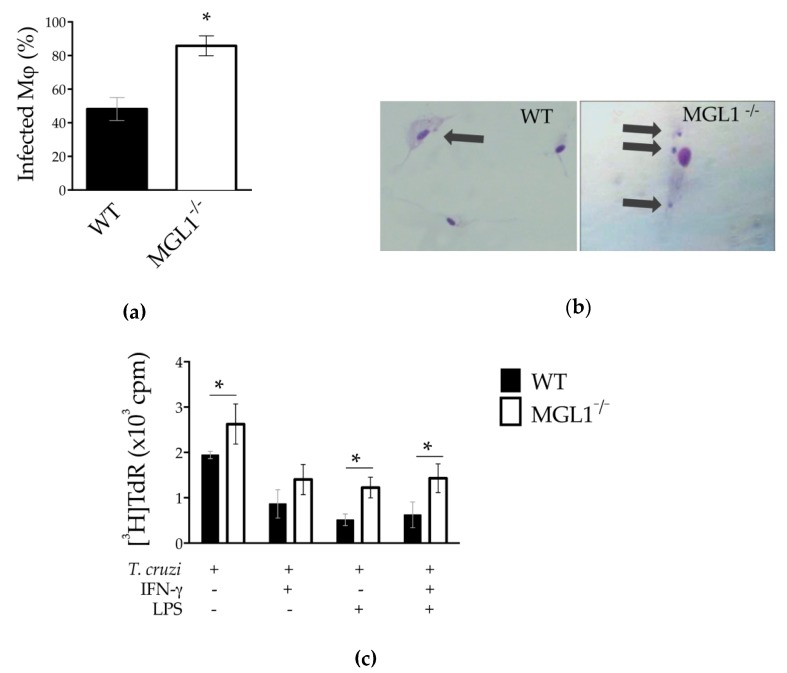
MGL1^−/−^ PE-Mφ take up more *T. cruzi* parasites. (**a**) Percentage of PE-Mφ with internal parasites in two hours after infection. (**b**) A representative image of parasites internalized in PE-Mφ; WT and MGL1^−/−^ PE-Mφ infected with *T. cruzi* (arrows point to parasites), magnification 40×. (**c**) Mφ from MGL1^−/−^ and WT mice not treated or treated overnight with IFN-γ, LPS or IFN-γ/LPS followed by exposure to *T. cruzi* at a 10:1 parasite/Mφ ratio. Parasites that were not taken up were removed at 2 h post-infection, and parasite infection was determined by lysis of the Mφ and the measurement of parasite proliferation by [^3^H]thymidine incorporation. The results are shown as the means of replicate samples (± SEM) and are representative of three experiments; * *p* < 0.05.

**Figure 3 cells-09-00108-f003:**
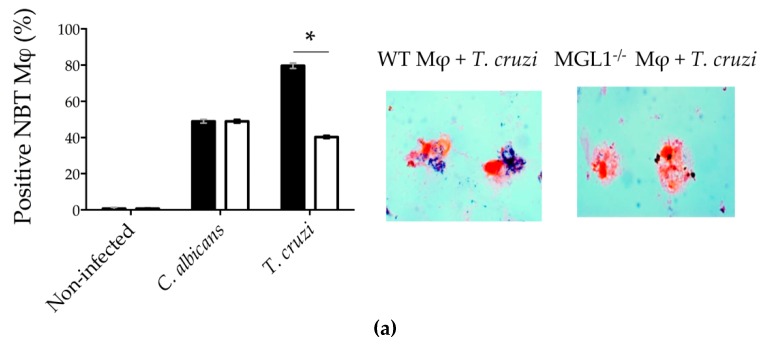
MGL1^−/−^ PE-Mφ have a deficient ROS, as well as NO and proinflammatory cytokine production during *T. cruzi* infection. (**a**) WT and MGL1^−/−^ PE-Mφ were infected for 2 h with culture-derived epimastigotes of *T. cruzi* or *Candida albicans* (as positive control), and ROS production was analyzed. (**b**) PE-Mφ from MGL1^−/−^ and WT mice were not treated or treated for 24 h with IFN-γ, LPS or IFN-γ/LPS followed by infection with epimastigotes of *T. cruzi* at a 10:1 parasite/Mφ ratio. Non-internalized parasites were removed at 2 h post-infection; supernatants were taken for NO and (**c**) cytokine IL-12, TNF-α and IL-10 quantification. The results are representative of three experiments; * *p* < 0.05.

**Figure 4 cells-09-00108-f004:**
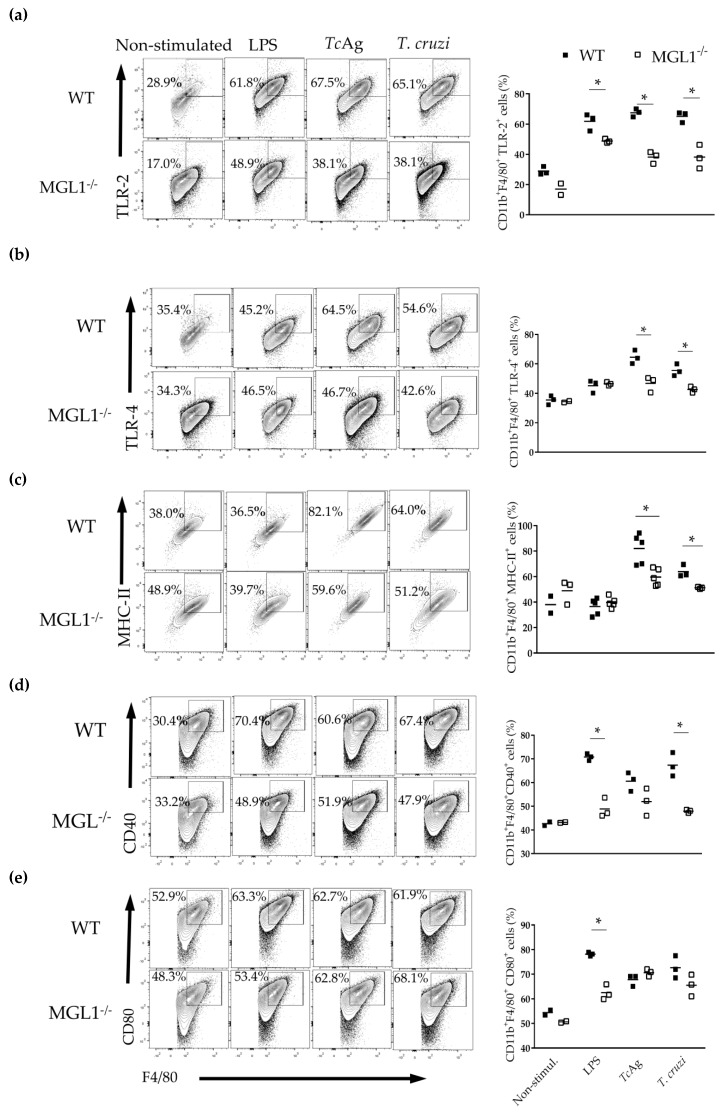
The absence of MGL1 results in reduced expression of TLR2, TLR4, MHCII and CD40 in PE-Mφ. (**a**–**e**) PE-Mφ from WT and MGL1^−/−^ mice were stimulated in vitro with LPS (100 ng/mL) or *Tc*Ag (25 µg/mL) or infected with epimastigotes of *T. cruzi* (ratio 1:10) for 24 h. The cells were stained with anti-F4/80, anti-TLR-2, anti-TLR-4, anti-MHC-II, anti-CD80 and anti-CD40. The dot plot and bar charts are representative of three independent experiments; * *p* < 0.05.

**Figure 5 cells-09-00108-f005:**
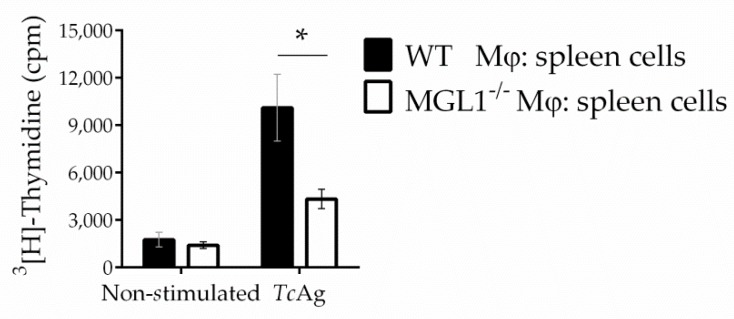
MGL1^−/−^ PE-Mφ induce a deficient activation of *T. cruzi* antigen-specific lymphocytes. PE-Mφ from MGL1^−/−^ or WT mice were stimulated with TcAg, two hours later Mφ were washed to remove the non-phagocytosed antigen. Splenocytes from 21 days-infected WT mice were added at ratio of 1:10. After 5 days, cell proliferation was assessed by [^3^H]thymidine incorporation. Data are representative of three separate experiments and are plotted as the means of triplicate wells (± SEM), n = 7, * *p* < 0.05.

**Figure 6 cells-09-00108-f006:**
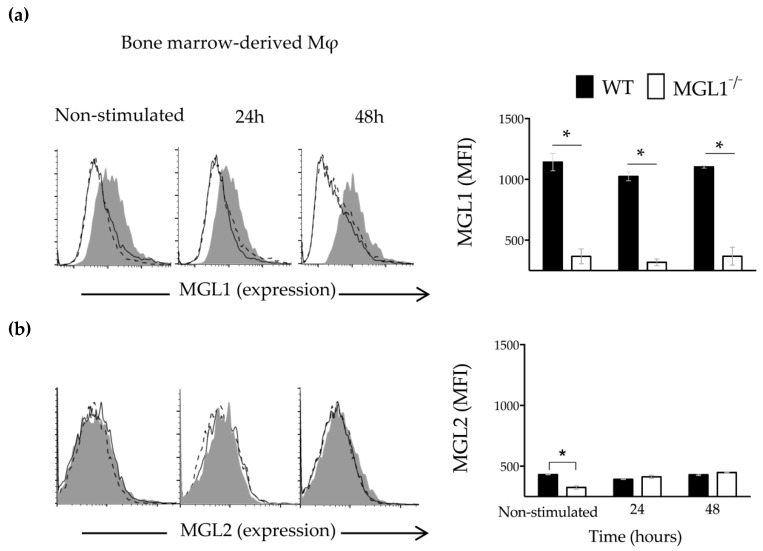
Expression of MGL1 and MGL2 in BMMφ in response to *TcAg*. BMMφ (F4/80+) from WT or MGL1^−/−^ mice were stimulated for 24 or 48 h with *Tc*Ag (25 µg/mL). Representative histogram and bar chart of the percentage of BMMφ expressing MGL1 and MGL2 are shown in (**a**) and (**b**), respectively. Dotted line, isotype; gray area, Mφ from WT mice; solid line, Mφ from MGL1^−/−^ mice; n = 6 mice per group; and * *p* < 0.05.

**Figure 7 cells-09-00108-f007:**
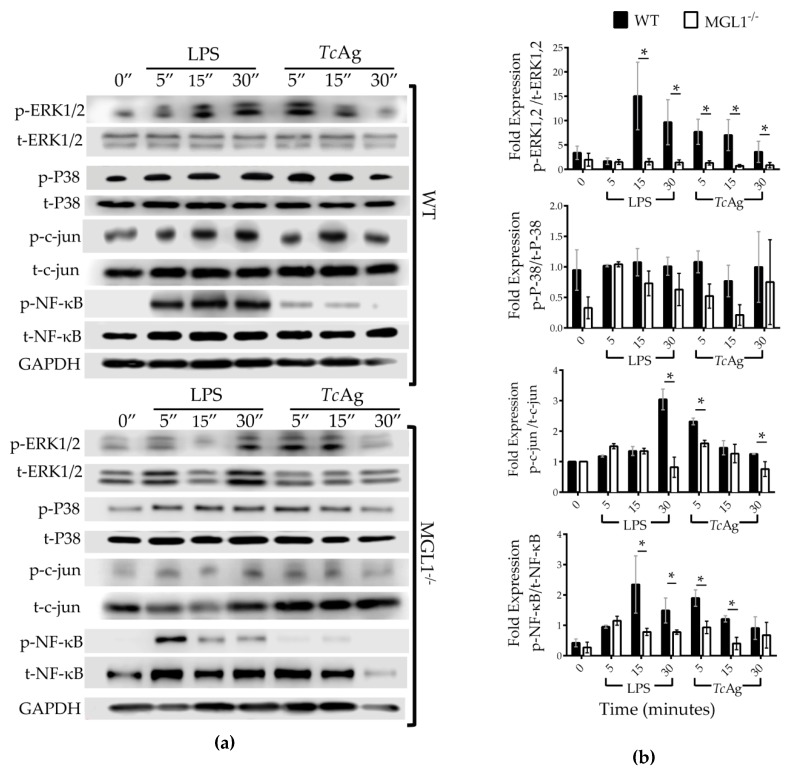
MGL1^−/−^ BMMφ exhibit reduced activation of ERK1/2, c-Jun and NFκ-B signaling pathway in response to *Tc*Ag. (**a**) WT and MGL1^−/−^ BMMφ were stimulated for 5, 15 and 30 min with LPS (100 ng/mL) or *Tc*Ag (25 µg/mL). In total protein lysates of Mφ were measured the phosphorylation of ERK1/2, p38 and NFκB by western blot analysis. Western blotting for the indicated proteins. (**b**) Densitometry analysis for the indicated proteins as described above. Results are representative of at least three separate experiments each with three biological replicates. Densitometry analysis of the indicated proteins was normalized first, to GAPDH protein, and next to total protein expression as appropriate. Data shown are mean ± SEM (n = 6), * *p* < 0.05.

**Figure 8 cells-09-00108-f008:**
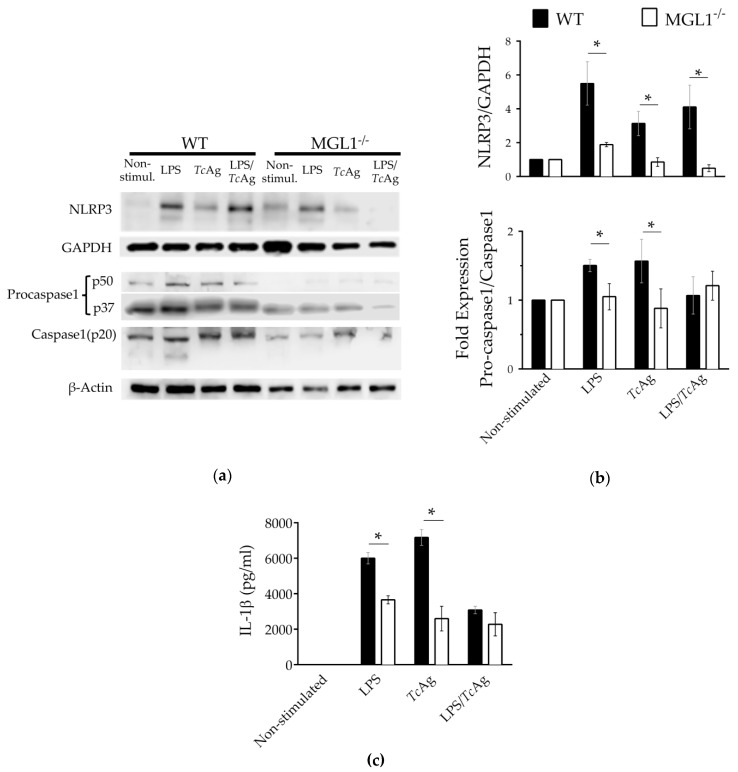
MGL1 is required for proper expression of NLRP3, pro-caspase1/caspase1 and IL-1β production. WT and MGL1^−/−^ BMMφ were stimulated for 24 h with LPS (100 ng/mL) or *Tc*Ag (25 µg/mL) or LPS+*Tc*Ag (100 ng + 25 µg/ mL), in total protein lysates of Mφ were measured NLRP3, pro-caspase1 and caspase1 by western blot. (**a**) Western blot showing down-regulated NLRP3, pro-caspase1 and caspase1 protein expression in MGL1^−/−^ Mφ, and (**b**) the densitometry analysis for the NLRP3 and the fold expression of pro-caspase1/caspase 1. (**c**) ELISA quantification of IL-1β in the cell culture supernatants. The western blot data were normalized to GAPDH control and are representative of two separate experiments. The data are plotted as the means (± SEM), n = 5, * *p* < 0.05.

**Figure 9 cells-09-00108-f009:**
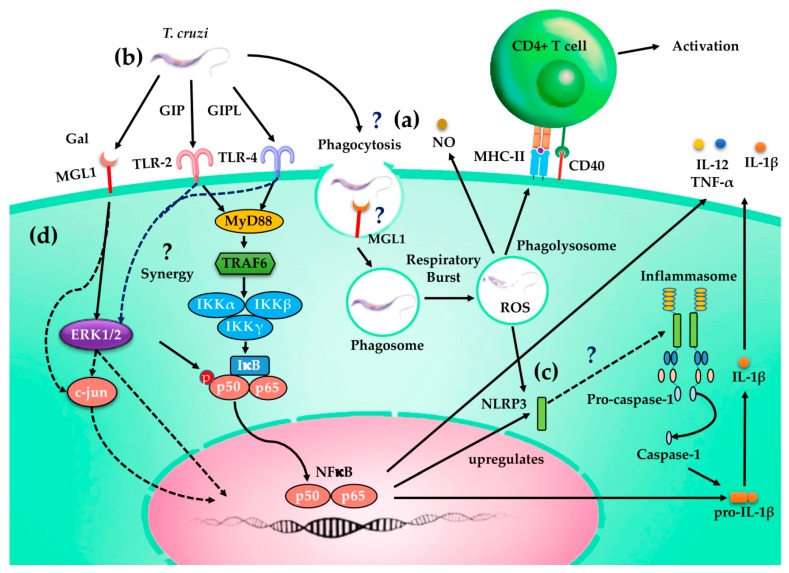
Functions of MGL1 in macrophage response to *T. cruzi* infection. (**a**) MGL1 recognizes glycosylated molecules and may mediate the uptake of *T. cruzi* through phagocytosis, inducing antimicrobial effector mechanisms such as respiratory burst and NO production. This leads to the induction of MHC-II and co-stimulatory molecules and increased antigen presentation to CD4+T cells. (**b**) The recognition of extracellular *T. cruzi* by MGL1 in Mφ induces intracellular signaling through the activation of ERK1/2 and NFκB, as well as the expression of TLR2 and TLR4, that result in the transcription of proinflammatory cytokines, such as TNF-α and IL-12, although anti-inflammatory cytokines such as IL-10 are also produced. (**c**) Moreover, MGL1 appears to favor the active form of the NLPR3 inflammasome; consequently, the assembled of pro-caspase 1, which in turn, by autoproteolysis, generates caspase 1 responsible for the cleavage pro-IL-1β to its active form of IL-1β. d) Dashed lines indicate incomplete understood mechanisms.

**Table 1 cells-09-00108-t001:** The numbers of Mφ that exhibit at least one amastigote, and the number of intracellular amastigotes per Mφ.

		Number of Amastigotes Per Macrophage				
Mφ Infected/100 cells		1	2–3	4	5	≥6
WT	48%	29%	14%	4%	1%	0%
MGL1^−/−^	86%*	22%*	33%*	21%*	7%*	3%

Values of * *p* < 0.05 were considered statistically significant compared to WT Mφ. These are representative of three independent experiments.

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
