# Peer review of "MGL1 Receptor Plays a Key Role in the Control of T. cruzi Infection by Increasing Macrophage Activation through Modulation of ERK1/2, c-Jun, NF-κB and NLRP3 Pathways"

_cells, 2020, doi:10.3390/cells9010108_

Round 1
Reviewer 1 Report
After reading the answers to my previous concerns, the three first have been clarified. However, the data now included to address my fourth concern need further clarification.
New data included in figure 8 are not properly interpreted in my opinion.
My interpretation would be that the expression of the overall inflammasome machinery is dampened in the Mgl1-deficient macrophages.
In fact, if we pay attention to the fold expression of Pro-caspase1/Caspase1, this value is reduced in Mgl1-/- cells, meaning that there is a higher proportion of active caspase; this would contradict the results shown in figure 8C.
I would recommend to monitor the expression of NLRP3, caspase-1 and IL1ß at the transcriptional level, because to the light of the provided results, this would be the most probable source for the reduced mature IL-1ß expression.
Please, change figure legend 8 according to the new data included.
In addition, the abstract should be reformulated according to those results, as the “activation of the inflammasome NLRP3” has not been addressed, but its expression. Of note, talk about the activation of procaspase 1 is not really accurate. Eventually, I think that the results are indicative of a reduced expression of all those components.
Depending on the results obtained from the experiment I recommend, the hypothetical model proposed in Figure 9 might need some reformulation.
Reviewer 2 Report
The paper from Rodriguez et al. show that MGL1 present in macrophages has an important role in defining the macrophage activation state during Trypanosoma cruzi infection. They have previously demonstrated that MGL1 KO mice are highly susceptible to T. cruzi infection and develop higher parasitemia and mortality rates than WT mice. Here, they propose that MGL1 has an unappreciated role in macrophage activation through the modulation of different signaling pathways involving ERK, c-Jun, NF-kB and NLRP3, and to the development of protective innate immunity against experimental T. cruzi infection.
The paper is acceptable, but some concerns need to be corrected and clarified in the manuscript before publication:
General: Please, use in the text all introduced abbreviations
- The definition of PECs is missing (M&M, lane 133).
-Page 3, lane 91: replace (NLRP)3 by NLRP3
- PE-Mφ have been defined in M&M, please use the abbreviation throughout the Ms: as example in Figure 1 legend (lane 267, resident peritoneal), Figure 2 legend (lane 298, peritoneal exudate macrophages), Figure 3 legend (lane 346, peritoneal exudate macrophages), Figure 5 legend (lane 375, peritoneal exudate macrophages), M&M (2.5, lane 149; 2.6, lane 159, peritoneal exudate macrophages), M&M (2.8, lane 195, peritoneal macrophages), etc.
-Figure 7 and Figure 8: Use BMMφ abbreviation.
M&M
-Please explain the rationale for the use of culture-derived epimastigotes to infect the cells in vitro, taking to account that infective form of T. cruzi is the trypomastigote. Do you expect that the few trypomastigote forms present in epimastigote cultures to be those that infect the cells?
-Do the authors think that during the culture of macrophages with epimastigotes, the parasites infect the cells (the few trypomastigotes present in the cultures)? Or do they think that the epimastigotes are phagocyted by the macrophages? The authors state in lane 183 that “ …to remove the non-phagocytosed parasites.”
- The use of 3H-Thymidine to evaluate the culture epimastigote proliferation is a very known method. On the other hand, an amastigote is also a replicative form of T. cruzi, but only intracellular. So, I am asking about the methodology described in this work to evaluate amastigote proliferation with 3H-Thymidine using a cell-free culture medium. Are you measuring thymidine incorporation (proliferation) of the amastigote form? Or, you expect the amastigotes recovered from the cells become epimastigotes in the 72 hours of culture before placing 3H-thymidine? Please, explain the rationale of these experiments.
Results
-Lane 256: the authors state that WT PE Mo have a low expression of MGL2 at baseline. I think that considering the MFI of the control isotype histogram, it is not correct to say that these cells express MGL2.
-Figure 1: Please, consider showing the statistical analysis between the different times post-infection (0h vs 24h; 24h vs 48h; in order to affirm that “After stimulation with TcAg for 24 or 48 h, MGL1 expression increased significantly while MGL2 expression increased moderately (Fig 1a, b; WT-Mφ, gray shadow).”
-In Figure 6, the authors show that BMMφ can up-regulate MGL1 but not MGL2 expression after TcAg stimulation. However, using these results they conclude that “These observations indicate that recognition and uptake of TcAg is mediated by MGL1 and does not by MGL2 in BMMφ.“ (lane 190). I think that the upregulation of MGL1 expression after antigen stimulation only indicates up-regulation after the recognition, and do not indicate that MGL1 participate in the uptake of antigen. Please, consider revising this conclusion.
-The following conclusion “These results demonstrate that BMMφ from MGL1-/- mice are unable to upregulate MGL1 or MGL2 in response to TcAg” is it is too obvious in relation to MLG1. Please, consider reformulating the conclusion including only MGL2.
Discussion
-Please, insert the bibliographic reference at the end of lane 482: “…. expression of MHCII and TLR4” ().
-In lane 479 the authors indicate that the dramatic downregulation of ROS and NO production as well as IL-12 and TNF secretion observed in macrophages MGL1 -/- after parasite infection suggest that MGL1 function would be to help to transport it and guide inside the cell………. . I think that there are not enough results to suggest that. Instead, I think that they have shown important bibliographic evidence that can support a cross-talk between MGL1 signaling pathway and other important pathways for macrophage activation (Napoletano et al.). In addition, the authors state this concept in lane 519: “Thus our results reveal….
-In this paper, the authors have demonstrated that macrophages MGL1 -/- infected in vitro with T. cruzi presented more internalized parasites than Wt ones. In addition, in lane 472 the authors state that. So, I can understand why in Figure 9 they state that MGL1 mediates the uptake of T. cruzi.
-
Author Response
Please see the attachment

This manuscript is a resubmission of an earlier submission. The following is a list of the peer review reports and author responses from that submission.
Round 1
Reviewer 1 Report
Authors and title:Tonathiu Rodriguez, Thalia Pacheco-Fernández, Alicia Vázquez-Mendoza, Imelda Juárez-Avelar, José Luis Reyes, Luis I. Terrazas, and Miriam Rodriguez-Sosa: MGL Receptor pays a key role in the control of Trypanosoma cruzi infection by increasing macrophage activation through modulation of NFκB, ERK1,2 and NLRP3 inflammasome pathways
Outline:Bone marrow-derived macrophages from MGL1/2-double knockout mice were used to evaluate the roles of MGL1, MGL2, or both in Trypanosome cruziinfection in vitrofocusing on the process of activation of NF-kB pathways. The results indicated that MGL1, MGL2, or both apparently acted to modulate activation signals in macrophages.
Major Critique:According to previous publications, mouse gene has MGL1 and MGL2, which have different amino acid sequence in the carbohydrate recognition domain and in the cytoplasmic domain. Also, their expression profiles are diverse among different subsets of macrophages and dendritic cells. It has been reported that MGL2 expression was rather limited to dendritic cells but the authors confirmed in this report that both MGL1 and MGL2 are expressed on bone marrow-derived macrophages used in this study. As the consequence, the results are confusing because it is not shown whether MGL1, MGL2, or both are involved in the modulation of macrophage activation and which is the macrophage subset involved in the pathogenesis of T. cruziinfection. Therefore, this seems to be an unfortunate situation in which the significant efforts shown in this report do not bear scientific merit beneficial to the area or science in general.
Conclusions:This reviewer judges that this report will become publishable when additional in vivostudies with MGL1/2-double KO mice is completed to assess how T. cruziinfection is affected, which macrophage subset is involved with the outcome, and which of MGL1, MGL2, or both are involved with the process
Reviewer 2 Report
In this manuscript, Rodriguez and colleagues address the signaling pathway responsible for the inflammatory responses triggered by MGL receptor in response to Trypanosoma cruzi. Some data are interesting and represent a new conceptual step such as the relevance of this receptor for adaptive antigen-specific responses. However, some of the provided data are not really new as the same group published some of them already (Vazquez, et al., Int. J. Biol. Sci., 2014). In addition, in some of the analyzed effects there is also an effect in response to LPS and/or IFNgamma, what complicate the interpretation of some results. Here are my concerns.
MAJOR CONCERNS
1. There is less NO production in MGL-deficient macrophages in response to LPS or LPS/IFNg without parasite. How do they explain this? In addition, having NO production only with these stimuli, it is difficult to know whether there is a contribution on NO production of the parasite upon those experimental conditions.
2. Figure 3C. Considering data on NO production, control conditions with only LPS or IFNg are missing. Is there cytokine production in response to those stimuli and if so, is it affected by the lack of MGL?. Effect to LPS stimulation on cytokine production in the absence of MGL is shown in (Vazquez, et al., Int. J. Biol. Sci., 2014).
3. For those parameters with effect also in response to LPS, does it reflect a “general” pro-inflammatory response of MGL-deficient macrophages? A negative control is required, showing no defects. Candida stimulation could be an option, for instance, as used in 3A.
4. NLRP3 expression is not indicative of inflammasome activation. In order to properly address inflammasome activation, and IL-1b processing, western blot of pro-IL1b and mature IL1b are required. In addition, based on the provide data, it cannot be ruled out an effect at the transcriptomic level (as NF-kB activation is dampened). Consequently, in order to claim about inflammasome activation, these determinations are required.